



# A global fire emission dataset using the three-corner hat method (FiTCH)

Meng Liu[1] and Linqing Yang[1]

[1]School of Biological Sciences, University of Utah, Salt Lake City, UT 84112, USA

*Correspondence to*: Meng Liu (meng.liu@tamu.edu) and Linqing Yang (linqingyang_bnu@tamu.edu)

**Abstract.** Fire carbon emissions contribute to the accumulation of atmospheric CO2 and affect climate change. It is crucial to accurately monitor the dynamics of global fire emissions for fire management and climate change mitigation. However, there are large uncertainties in the existing satellite-based global fire emission products. This study analyzed the uncertainties of six state-of-the-art fire emission products and merged them using the three-corner hat method (TCH),
producing a new global fire emission dataset, FiTCH. Our results revealed that satellite-based products such as the Global Fire Assimilation System (GFAS), the Quick Fire Emissions Dataset (QFED), and the Global Fire Emissions Database (GFED) had low uncertainties in fire emissions, while the Fire INventory from National Center for Atmospheric Research (NCAR) (FINN), the Fire Energetics and Emissions Research (FEER), and Xu et al. (2021) data had high uncertainties. The proposed FiTCH dataset presented the lowest uncertainties with a mean annual fire emission of 1978.47 Tg C in 2001–2021.
Among biomes, tropical forests and tundra showed higher uncertainties than other biomes such as temperate forests and Mediterranean forests. In drought years, forests showed increased fire emissions, especially in boreal forests, while non-forest regions like grasslands displayed decreased emissions. By integrating the FiTCH data and historical fire emissions in the late 20th century, 1994 was identified as a break year, before which global fire emissions increased significantly and after which the emissions decreased. Global land temperatures and fire emissions have decoupled in the past two decades.
However, climate change still causes threats to forest carbon sequestration, especially for boreal forests. This study highlights the importance of forest fire monitoring and management for effective climate mitigation and ecosystem conservation. The proposed FiTCH dataset is available from: https://doi.org/10.6084/m9.figshare.22647382.v1 (Liu and Yang, 2023).

## 1 Introduction

Fire, as an important natural disturbance, impacts land surface phenology and mediates ecosystem functioning and biogeochemical cycling (Bowman et al., 2020; Walker et al., 2019; Wang and Zhang, 2020). Biomass burning during fire events releases sequestered carbon, alters atmosphere composition, and decreases land carbon storage. The mean annual global fire carbon emissions were ~2 Pg C in 1997–2016 (van der Werf et al., 2017), approximately 21% of annual fossil fuel emissions (Friedlingstein et al., 2022). Fire related activities such as deforestation and peatland burning are usually

irreversible and become net carbon sources. In general, about 24% of global fire emissions are net carbon sources and contribute to atmospheric CO2 accumulation and global warming (Bowman et al., 2009; van Wees et al., 2022). The change and dynamics of fire emissions are pivotal to the global carbon cycle. It is critical to figure out the dynamics of global fire emissions for ecosystem conservation and climate change mitigation.

Satellite observations have been a consistent and reliable data source for global fire monitoring and fire emission estimation. There are three broad approaches for using satellite data to estimate fire emissions. First, the bottom-up approach uses burned area and fuel loads to estimate burned biomass and carbon emissions (Seiler and Crutzen, 1980; van der Werf et al., 2017). The burned biomass is the product of the burned area, fuel loads, and combustion completeness, and the fire emissions are usually obtained by multiplying the emission factor and the burned biomass (van der Werf et al., 2017). The

Global Fire Emissions Database (GFED) (van der Werf et al., 2017) was produced using the bottom-up approach, where the burned area and fuel loads (leaves, stems, coarse woody debris, and litter) came from the Moderate Resolution Imaging Spectroradiometer (MODIS) data and the Carnegie–Ames–Stanford Approach (CASA), respectively. Second, the fire emissions can be estimated based on fire radiative power (FRP), which is the radiative energy released by biomass combustion. As the integrated FRP is linearly correlated with the consumed biomass (Wooster, 2002; Wooster et al., 2005),

the burned biomass and the corresponding emissions can be calculated accurately. The Global Fire Assimilation System (GFAS) product (Kaiser et al., 2012) was produced based on the MODIS FRP data. Third, spaceborne light detection and ranging (LiDAR) data can be used to estimate biomass storage and fire induced biomass change (Liu and Popescu, 2022; Xu et al., 2021). Forest biomass was estimated using LiDAR derived tree height and canopy structure, and the consumed biomass was defined as the biomass change due to fire events. Xu et al. (2021) estimated global fire emissions from 2001 to

2019 based on the combination of the LiDAR data from the Ice, Cloud, and land Elevation Satellite (ICESat) and the optical data from MODIS.

However, there are considerable uncertainties in the current satellite-based fire emission products. Stroppiana et al. (2010) compared global CO emissions from five emission products and revealed that CO emissions in 2003 ranged between 365 to

1422 Tg. Carter et al. (2020) analyzed four biomass burning aerosol datasets over North America and found that the emissions differed by a factor of 4 to 7. The large uncertainties are related to the limitations in the fire emission estimation methods. For instance, the performance of the bottom-up approach is subjected to the accuracy of the burned area, fuel loads, fuel composition, and fuel moisture content. The GFED was reported to show an uncertainty of ±50% in continental fire emission estimation (van der Werf et al., 2017; van Wees et al., 2022). For the FRP based method, variations in detection

efficiency, cloud and smoke obstructions, and dissipation of fire radiative energy can cause challenges to fire emission monitoring (Freeborn et al., 2011; Randerson et al., 2012). For the LiDAR based method, the estimation of fire emissions in boreal regions is not straightforward because the belowground biomass consumed by boreal forest fires is difficult to



measure directly using LiDAR data. Additionally, optical data are still indispensable when producing wall-to-wall biomass maps as the footprints of spaceborne LiDAR are usually sparse (Saatchi et al., 2011; Xu et al., 2021).


This study aims at illuminating the uncertainties of the state-of-the-art fire carbon emission products and producing a new global fire emission dataset with low uncertainty and high reliability. We employed six widely used satellite-based global fire emission products, including the GFAS (Kaiser et al., 2012), the GFED (van der Werf et al., 2017), the Fire INventory from National Center for Atmospheric Research (NCAR) (FINN) (Wiedinmyer et al., 2023, 2011), the Fire Energetics and

Emissions Research (FEER) (Ichoku and Ellison, 2014), the Quick Fire Emissions Dataset (QFED) (Koster et al., 2015), and the global fire emission dataset from Xu et al. (2021). The three-corner hat (TCH) method (Premoli and Tavella, 1993) was leveraged to derive the uncertainties and merge the six fire emission products, producing a new global fire emission dataset, FiTCH. The uncertainties of different products were compared, and the dynamics of global fire emissions in the past two decades were investigated. Fire emissions in different regions and different biomes were derived and analyzed. The effects of

climate extremes, i.e., drought, on fire emissions were also evaluated. Our primary goals were 1) to illustrate the uncertainties of different fire emission products, 2) to produce a new global fire emission dataset with low uncertainty, and 3) to figure out the climate effects on fire emissions.

## 2 Materials and methods

### 2.1 Data

#### 2.1.1 Satellite-based fire emission products

Six state-of-the-art satellite-based fire emission products were employed in this study. The GFAS (version 1.2) (Kaiser et al., 2012) used the MODIS FRP to estimate global 0.1° fire emissions since 2003. The FRP and biome specific conversion factor (the biomass consumed per megajoule) were multiplied to estimate the burned biomass and derive fire emissions. The GFAS emission data in 2003–2021 were downloaded and analyzed in this study. The GFED (version 4s) (van der Werf et al., 2017)

combined satellite observed burned area and the CASA derived fuel loads to evaluate burned biomass and fire emissions. The GFED data in 2001–2021, whose original spatial resolution was 0.25°, were resized to 0.1° and employed in this study. The FINN (version 2.5) (Wiedinmyer et al., 2023) fire emissions were based on MODIS active fire detection, land cover types, and corresponding fuel consumption, producing a global 0.1° emission product in 2002–2021. The FEER (version 1.0 G1.2) (Ichoku and Ellison, 2014) data were based on the total particulate matter (TPM), which was a product of the FRP and

the estimated emission coefficients. The TPM was converted to different components using emission factors from Andreae and Merlet (2001), producing a global 0.1° fire emission product since 2003. The FEER data in 2003–2021 were used in this study. The QFED (version 2.5) (Koster et al., 2015) was based on the FRP and drew on the cloud correction method in the GFAS to produce global 0.1° fire emissions since 2001, in which the emission data in 2001–2021 were employed in this



study. The spaceborne LiDAR data derived fire emissions in Xu et al. (2021) were produced with LiDAR based plant
biomass and MODIS burned area, providing global 0.1° annual fire emissions in 2001–2019.

### 2.1.2 Biomes

The global biome map (Olson and Dinerstein, 2002), which covered 16 biomes defined by the World Wildlife Fund (WFF),
was employed in this study. Inland water and Rock and ice biomes were removed, and the rest 14 biomes were merged into
seven major types: Grasslands, savannas and shrublands (GRASS), Tropical and subtropical forests (TROPICS), Boreal
forests (Boreal), Temperate forests (Temperate), Desert and xeric shrublands (DEXS), Mediterranean forests, woodlands and
scrub (MFWS), and Tundra. The merging rules are displayed in Table 1, and the biome map is shown in Fig. 2a.

**Table 1**. The original 14 biomes and the merged seven major biomes.

| The original biomes | The aggregated biomes | Abbreviations |
|---|---|---|
| Tropical and Subtropical Grasslands, Savannas and Shrublands | Grasslands, savannas, and shrublands | GRASS |
| Temperate Grasslands, Savannas and Shrublands | | |
| Montane Grasslands and Shrublands | | |
| Flooded Grasslands and Savannas | | |
| Tropical and Subtropical Moist Broadleaf Forests | Tropical and subtropical forests | TROPICS |
| Tropical and Subtropical Dry Broadleaf Forests | | |
| Tropical and Subtropical Coniferous Forests | | |
| Mangroves | | |
| Boreal Forests/Taiga | Boreal forests | Boreal |
| Temperate Broadleaf and Mixed Forests | Temperate forests | Temperate |
| Temperate Conifer Forests | | |
| Deserts and Xeric Shrublands | Desert and xeric shrublands | DEXS |
| Mediterranean Forests, Woodlands and Scrub | Mediterranean forests, woodlands, and scrub | MFWS |
| Tundra | Tundra | Tundra |

### 2.1.3 Drought

Drought was identified using Palmer Drought Severity Index (PDSI) (Palmer, 1965), which was obtained from the
TerraClimate (Abatzoglou et al., 2018). Global 4 km monthly PDSI data since 1959 were downloaded and averaged to 0.1°
at the annual level. Generally, PDSI ranged from −6 to 6, where negative PDSI represented dry conditions and positive PDSI
indicated wet. In this study, a drought year was detected when the annual PDSI was lower than −2. For each 0.1° resolution
pixel, the average fire emissions in both drought years (PDSI < −2) and non-drought years (PDSI > −2) were calculated, and
the difference between them was derived.

### 2.1.4 Temperature

Global land temperature anomalies were downloaded from the National Ocean and Atmospheric Administration (NOAA)
(https://www.ncei.noaa.gov/access/monitoring/climate-at-a-glance/global/time-series), which provided annual temperature
data since 1880. The global land temperature anomalies were derived with respect to the 1901–2000 average. The unit of the
temperature data used in this study was degree Celsius (°C).



### 2.2 The three-corner hat (TCH) method

The TCH method (Premoli and Tavella, 1993; Xu et al., 2020) was developed to evaluate the uncertainties of different products when the true values were unknown. Using the uncertainties as weights, where high uncertainties led to low weights, a merged product could be produced. The six satellite-based fire emission products were combined in this study to create a merged fire carbon emission dataset using the TCH method (referred to as FiTCH hereafter).

In the TCH, an arbitrary product was used as the reference. For each $0.1°$ pixel, the differences between the rest $n - 1$ products and the reference were calculated,

$$D = \begin{bmatrix} d_{11} & d_{12} & \cdots & d_{1(n-1)} \\ d_{21} & d_{22} & \cdots & d_{2(n-1)} \\ \vdots & \vdots & \ddots & \vdots \\ d_{m1} & d_{m2} & \cdots & d_{m(n-1)} \end{bmatrix}, \tag{1}$$

where $d_{ij} = y_{ij} - y_i$ was the difference between the $j$th product at the $i$th year ($y_{ij}$) and the reference at the $i$th year ($y_i$), assuming there were $m$ years. The variance/covariance matrix of $D$ was represented by $S$,

$$S = \begin{bmatrix} s_{11} & s_{12} & \cdots & s_{1(n-1)} \\ s_{12} & s_{22} & \cdots & s_{2(n-1)} \\ \vdots & \vdots & \ddots & \vdots \\ s_{1(n-1)} & s_{2(n-1)} & \cdots & s_{(n-1)(n-1)} \end{bmatrix}, \tag{2}$$

where $s_{jk}$ was the covariance of $d_j$ (the $j$th column in the matrix $D$) and $d_k$ (the $k$th column in the matrix $D$). When $j = k$, $s_{jk}$ was the variance of $d_j$. The matrix $S$ could be written as the following equation (Premoli and Tavella, 1993; Xu et al., 2020),

$$S = KRK^{\mathrm{T}}$$

$$K = \begin{bmatrix} 1 & 0 & \cdots & 0 & -1 \\ 0 & 1 & \cdots & 0 & -1 \\ \vdots & \vdots & \ddots & \vdots & \vdots \\ 0 & 0 & \cdots & 1 & -1 \end{bmatrix}, \tag{3}$$

$$R = \begin{bmatrix} r_{11} & r_{12} & \cdots & r_{1n} \\ r_{12} & r_{22} & \cdots & r_{2n} \\ \vdots & \vdots & \ddots & \vdots \\ r_{1n} & r_{2n} & \cdots & r_{nn} \end{bmatrix}$$

where $K$ was a designed matrix ($(n - 1) \times n$) composed of an identity matrix ($I$) and a column vector ($-\mathbf{1}$, whose elements were all $-1$). $R$ was the variance/covariance matrix of errors, where $r_{jk}$ represented the covariance of errors of the $j$th product and the $k$th product. As the true value was unknown, $r_{jk}$ was also unidentified. Based on Eq. (3), $r_{jk}$ could be written as





$$r_{jk} = s_{jk} - r_{nn} + r_{jn} + r_{kn},\qquad(4)$$

where only $r_{nn}$, $r_{jn}$, and $r_{kn}$ were required in order to determine $r_{jk}$ because $s_{jk}$ was already known. In this way, the $(n-1)\times(n-1)$ subset of the matrix $R$ could be written as

$$R_{[1:(n-1),\,1:(n-1)]} = S - r_{nn}\left[uu^{\mathrm{T}}\right] + ru^{\mathrm{T}} + ur^{\mathrm{T}},\qquad(5)$$

where $u$ was a column vector (**1**, whose elements were all 1). $r$ was also a column vector whose elements were $r_{1n}$, $r_{2n}$, …, $r_{(n-1)n}$. When $r$ and $r_{nn}$ were figured out, the whole matrix $R$ could be calculated. The diagonal elements of the matrix $R$, $r_{jj}$, were the variances of errors of the corresponding products.

Based on the Kuhn–Tucker theorem (Galindo and Palacio, 1999), the quadratic mean of covariances ($r_{jk}$, where $j \neq k$) in the matrix $R$ should be minimized.

$$J = \sum_{j<k}\frac{r_{jk}^{2}}{L^{2}},\ s.t.\ -\frac{r_{nn} - \left[r - r_{nn}u\right]^{\mathrm{T}} S^{-1}\left[r - r_{nn}u\right]}{L} < 0,\qquad(6)$$

where $L = \sqrt[n-1]{\det(S)}$, and $r$ was included when calculating $J$. The initial values of $r$ were usually set to $r_{jn} = 0$ and $r_{nn} = (2u^{\mathrm{T}}S^{-1}u)^{-1}$. By finding the minimum $J$, the best $r_{jn}$ and $r_{nn}$ were derived and the matrix $R$ could be figured out. The diagonal elements of the matrix $R$, which represented the variances of errors, were used to merge the six fire emission products,

$$\hat{y}_i = \sum_{j=1}^{n} y_{ij} * w_j,\ w_j = \frac{1}{r_{jj}}\bigg/\sum_{j}^{n}\frac{1}{r_{jj}},\qquad(7)$$

where $w_j$ was the weight of the $j$th product, and $\hat{y}_i$ was the TCH estimated fire emission at the $i$th year. The variance of $\hat{y}_i$ was $1/\left(\sum\frac{1}{r_{jj}}\right)$. In this study, we used the shared years (2003–2019, $m = 17$) among the six fire emission products to train the TCH and derive the variances ($r_{jj}$) and weights ($w_j$). For 2003–2019, the FiTCH emissions were the weighted sum of the six satellite-based products. However, for 2020–2021, there were only five products (the GFAS, the GFED, the FINN, the FEER, and the QFED) available. Therefore, the FiTCH emissions in 2020–2021 were the weighted sum of the five products based on Eq. (7), where only five variances ($r_{jj}$) were used. For the rest years, the same procedures (remove $r_{jj}$ of the missing products and recalculate the weights for the remaining products) were employed to merge these products. The produced FiTCH global fire emission data were 0.1° from 2001 to 2021 at the annual scale. In this study, the uncertainties of fire emissions were represented by the standard deviations of errors, i.e., the square root of the variances of errors ($\sqrt{r_{jj}}$).

**2.3 Drought impacts on fire emissions**

For each 0.1° resolution pixel, the difference between average fire emissions in drought years (PDSI < −2) and non-drought years (PDSI > −2) were derived.

$$\Delta C = \bar{y}_{\mathrm{drought}} - \bar{y}_{\mathrm{non\text{-}drought}}\qquad(8)$$

where $\bar{y}_{\text{drought}}$ and $\bar{y}_{\text{non-drought}}$ were the average fire emissions in drought years and non-drought years, respectively. $\Delta C$ was the difference in the average fire emissions. When $\Delta C$ was greater than zero, drought contributed to producing more fire emissions. Conversely, drought suppressed fire emissions when the difference was lower than zero. For pixels without drought years or fire emissions, the differences were set to NA.

## 165 3 Results

### 3.1 Uncertainties in fire emissions

The uncertainties of fire emissions varied among the six satellite-based fire emission products and the proposed FiTCH dataset. The mean uncertainties (i.e., standard deviations of errors) were the smallest in the FiTCH data (mean = 0.78 Gg C) and the largest in the FINN data (mean = 4.99 Gg C) in Fig. 1. The mean uncertainties of the GFAS, the QFED, the GFED,
Xu et al. (2021), and the FEER data were 1.96, 2.89, 3.69, 4.83, and 4.92 Gg C, respectively. Tropical regions had high uncertainties in fire emissions, while other regions such as eastern North America, Europe, and western Asia had low uncertainties.

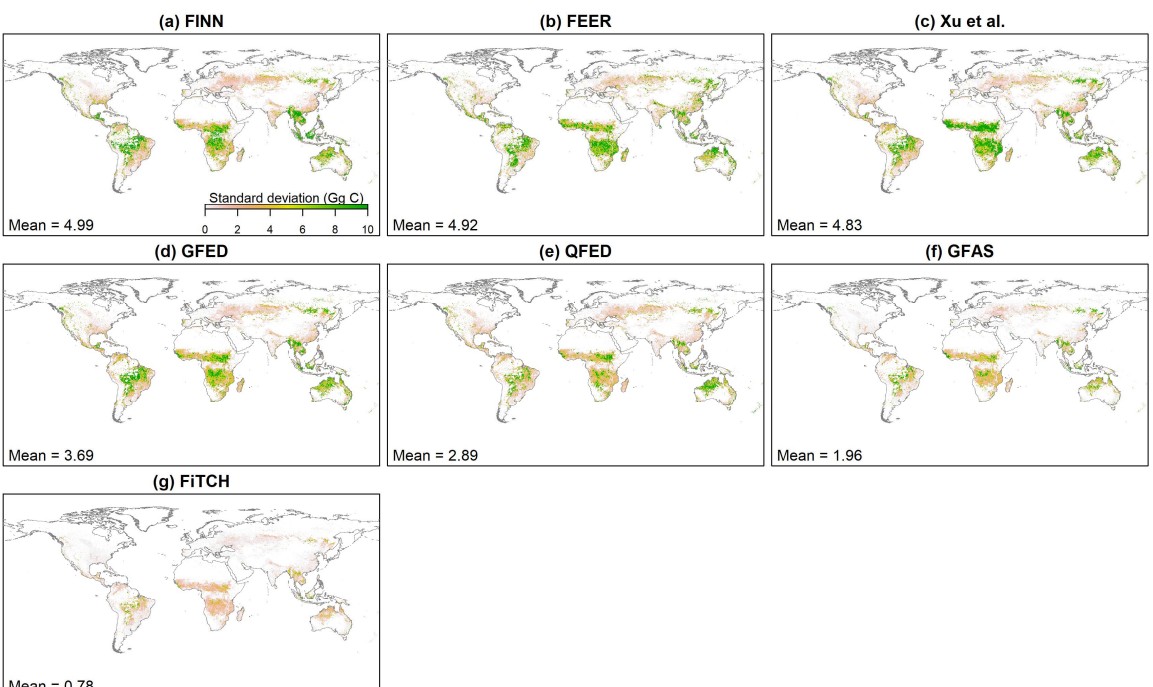

**Figure 1**. Uncertainties of the six satellite-based fire emission products and the proposed FiTCH dataset. The uncertainties were standard
deviations of errors (sqrt root of variances of errors). 1 Gg = $10^9$ g.
When considering different biomes (Fig. 2), Tropical and subtropical forests (TROPICS) and Tundra had high uncertainties, especially for the FINN and the FEER data. Temperate forests (Temperate) and Mediterranean forests, woodlands and scrub (MFWS) had low uncertainties. For Boreal forests (Boreal), the uncertainties of fire emissions in Xu et al. (2021) were the largest, 4.83 Gg C. For Desert and xeric shrublands (DEXS), the FEER data had the highest uncertainties (mean = 3.16 Gg C), which were slightly higher than those of the QFED (mean = 3.07 Gg C) and Xu et al. (2021) data (mean = 2.94 Gg C). For Grasslands, savannas, and shrublands (GRASS), Xu et al. (2021) had the largest uncertainties, 5.99 Gg C, and the FEER ranked second, 5.60 Gg C. For Mediterranean forests, woodlands and scrub (MFWS) and Temperate forests (Temperate), the FEER data had the largest uncertainties, 2.79 and 2.81 Gg C, respectively. The FINN and the FEER data had the largest uncertainties in TROPICS and Tundra, 10.23 and 8.31 Gg C, respectively. Other products such as Xu et al. (2021) and the GFED also had high uncertainties in TROPICS (mean > 5.1 Gg C) and Tundra (mean > 4.8 Gg C). The proposed FiTCH data had the lowest uncertainties in all biomes, in which TROPICS had the largest uncertainties, 1.10 Gg C.

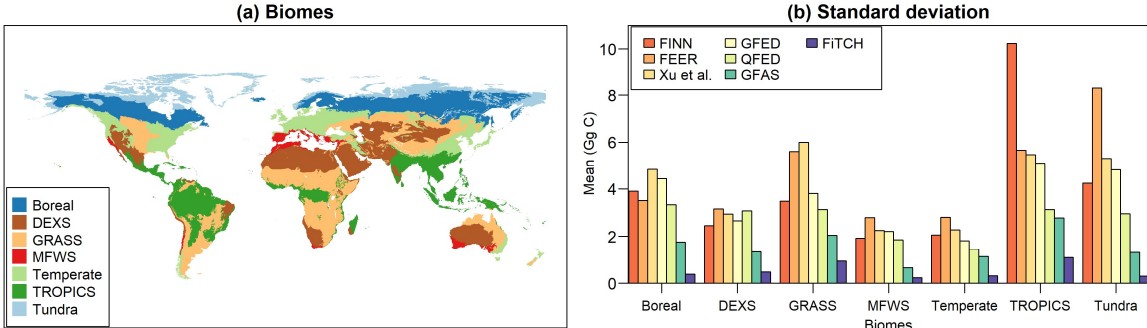

**Figure 2**. Uncertainties of fire emissions in different biomes: (a) the distribution of biomes and (b) the mean uncertainties of fire emissions. The uncertainties were standard deviations of errors (sqrt root of variances of errors). Seven major biomes were included: Boreal forests (Boreal), Desert and xeric shrublands (DEXS), Grasslands, savannas, and shrublands (GRASS), Mediterranean forests, woodlands and scrub (MFWS), Temperate forests (Temperate), Tropical and subtropical forests (TROPICS), and Tundra.

### 3.2 Trends of fire emissions

Global fire emissions decreased significantly in the past two decades but had large divergences. Among the six satellite-based products and the proposed FiTCH dataset, five of them presented significantly decreased trends. As shown in Fig. 3a, the FEER, Xu et al. (2021), the QFED, the GFAS, and the FiTCH exhibited significantly decreased fire emissions, whose slopes of regression between fire emissions and year were significantly lower than zero. The fire emission changes in the FINN and the GFED were not significant. Additionally, there were large differences among the annual fire emissions, where the FEER and the FINN data had high emissions while the QFED, the GFAS, the GFED, and the FiTCH showed low emissions. In recent years, e.g., in 2019 and 2021, the fire emissions were high compared with those in the 2010s. In Fig. 3b,

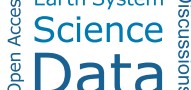

the FEER had the largest mean annual fire emissions, 3808.47 Tg C, and the FINN ranked second, 3198.87 Tg C. The mean annual fire emissions of the QFED (2186.77 Tg C), the GFED (2071.84 Tg C), the GFAS (2034.66 Tg C), and the FiTCH (1978.47 Tg C) were low and comparable. All the products presented decreased trends in fire emissions from −42.97 Tg C yr$^{-1}$ to −6.09 Tg C yr$^{-1}$, though the trends of the GFED and the FINN were not significant.

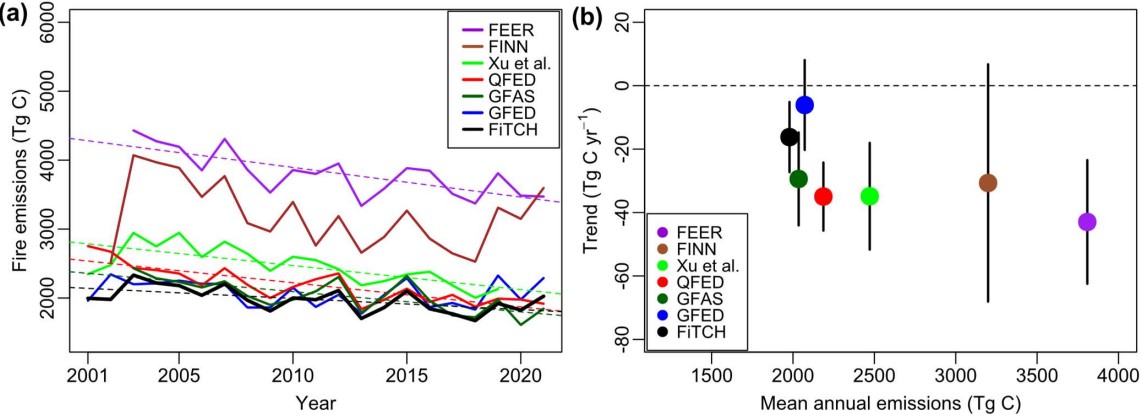

**Figure 3**. Trends of global fire emissions in the past two decades: (a) annual fire emissions, where the dashed lines are regression lines (only lines whose $p < 0.05$ are shown); (b) trend and mean annual emissions of each product, where the black lines are the 95% confidence interval. 1 Tg = 1000 Gg.

In most regions of the globe, the means of trends from the seven products were close to zero (Fig. 4a), indicating small changes in fire emissions. However, tropical regions in South America, Africa, and Asia presented large decreases in fire emissions. Other regions such as western North America and high latitudes in eastern Asia exhibited increased fire emissions. Overall, there were 54.21% of pixels presenting decreased fire emissions in the past two decades. In Fig. 4b, there were high standard deviations of fire emission trends in tropical regions, revealing large variation in fire emission estimation

over these regions. This result was consistent with the uncertainty distribution in Fig. 1, where tropical regions had high uncertainties in fire emissions.

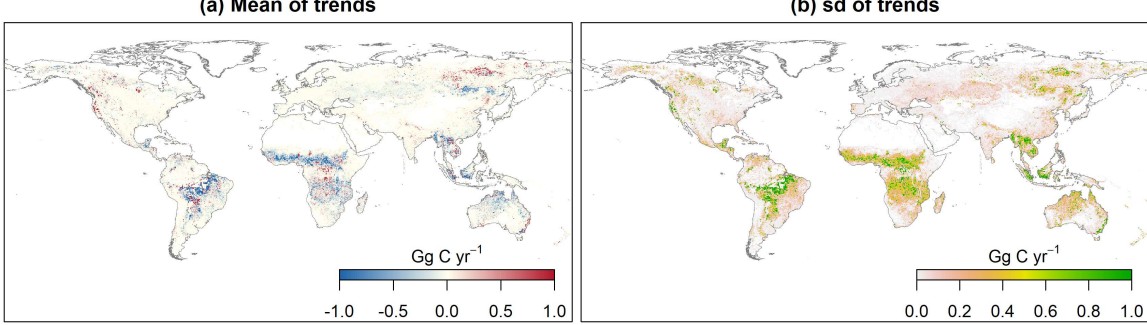



**Figure 4**. Variations of fire emission trends at the pixel level: (a) the mean of trends; (b) the standard deviation (sd) of trends from the seven fire emission products.

### 3.3 Spatial distribution of fire emissions

Global fire emissions exhibited considerable spatial heterogeneity. Based on the FiTCH dataset, tropical and subtropical regions such as South America, Africa, Southeast Asia, and northern Australia had high emissions, while fire emissions in the rest regions were low (Fig. 5a). There were also high emissions for certain places such as California, southeast Mexico, and eastern New South Wales. Other regions like eastern North America, Europe, and eastern Asia had very low emissions. Similar patterns were found in the six satellite-based fire emission products in Fig. S1. Notably, mean annual fire emissions in Xu et al. (2021) were low in the northern high latitudes (e.g., boreal regions) than the other products. In Fig. 5b, the emission trends of most pixels were close to zero, while tropical regions had large decreases in fire emissions. Overall, there were 52.96% of pixels revealing decreasing fire emission trends. The trends in Fig. 5b were significantly correlated with those in Fig. 4a, Spearman $r = 0.91$ and $p < 0.0001$.

**(a) Mean annual emissions**      **(b) Trend of emissions**

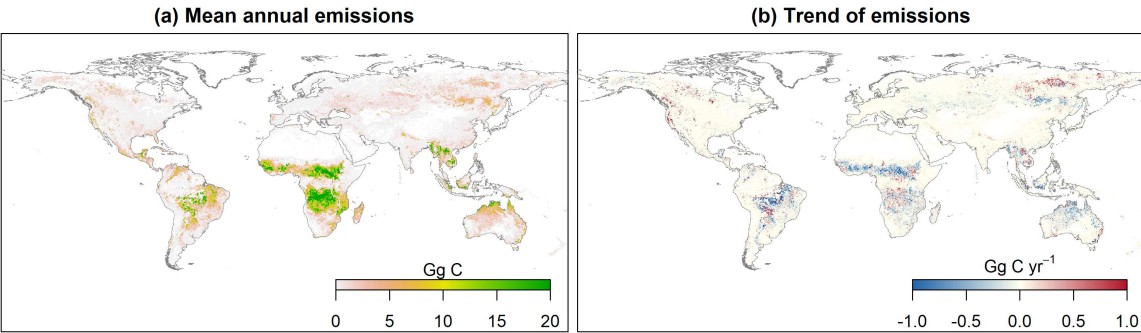

**Figure 5**. Global distribution of fire emissions based on the proposed FiTCH dataset: (a) the mean annual fire emissions; (b) the trend of fire emissions.

Grasslands, savannas, and shrublands (GRASS) had the highest mean annual fire emissions. In Fig. 6, mean annual fire emissions of the seven biomes ranged from 10.09 Tg C to 1100.47 Tg C, where Tundra and GRASS had the lowest and the highest emissions, respectively. The mean annual fire emissions in Tropical and subtropical forests (TROPICS) ranked second, 511.95 Tg C. Boreal forests exhibited significantly increased trends in fire emissions, 4.29 Tg C yr$^{-1}$. Temperate forests (Temperate), Mediterranean forests, woodlands and scrub (MFWS), and Tundra showed insignificant changes though their trends were positive. In GRASS, fire emissions decreased significantly, $-15.53$ Tg C yr$^{-1}$, in the past two decades. TROPICS and Desert and xeric shrublands (DEXS) had insignificant changes in fire emissions though the trends were





negative. These results provided a clue that the decrease in global fire emissions was mainly due to the reduced emissions in GRASS, TROPICS, and DEXS. The scatterplots of annual fire emissions of the seven biomes were presented in Fig. S2.

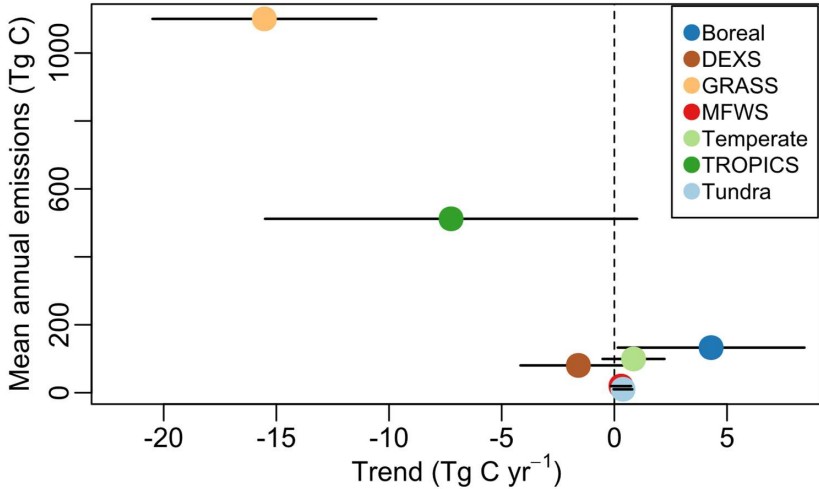

**Figure 6**. Fire emission trends and mean annual emissions in different biomes using the FiTCH dataset, where the black lines are the 95% confidence interval. 1 Tg = 1000 Gg.

### 3.4 Drought impacts

Drought contributed to increasing fire emissions in forests while suppressing fire emissions in non-forest regions. The difference between average fire emissions in drought years and non-drought years was calculated for each pixel. As shown in Fig. 7a, tropical and subtropical savannas in South America, Africa, and northern Australia presented decreased fire emissions in response to drought. In contrast, regions such as western North America, tropical forests in South America, tropical forests in Asia, and northern high latitudes in Asia exhibited increased emissions. The rest regions such as eastern North America, Europe, and eastern Asia had only slight changes, which were close to zero. Overall, 48.13% of pixels presented increased fire emissions in response to drought. In Fig. 7b, fire emissions increased significantly in five of the seven biomes due to drought, including Tundra, Tropical and subtropical forests (TROPICS), Temperate forests (Temperate), Mediterranean forests, woodlands and scrub (MFWS), and Boreal forests (Boreal). Boreal forests had the largest increases, 242.09 Tg C. TROPICS and Temperate forests ranked second and third, 112.73 and 76.30 Tg C, respectively. For Grasslands, savannas, and shrublands (GRASS) and Deserts and xeric shrublands (DEXS), fire emissions decreased significantly in drought years. In general, non-forest biomes like GRASS and DEXS tended to show decreased fire emissions in response to drought, expect Tundra.



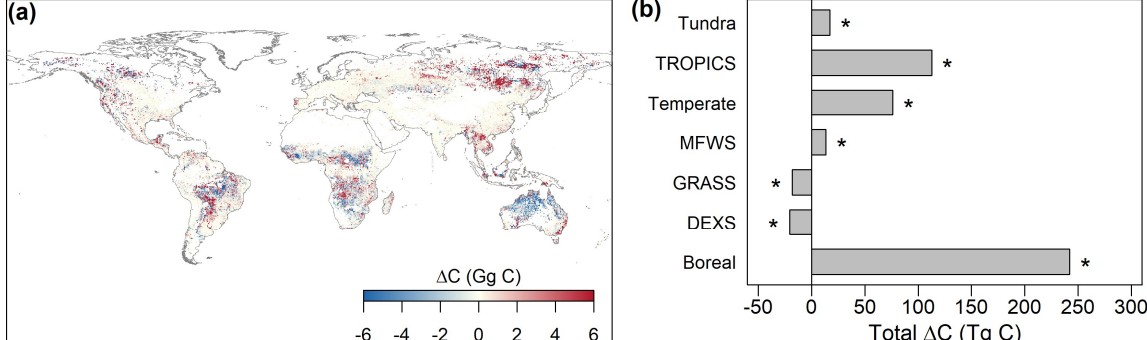

**Figure 7**. Fire emission change (ΔC) in response to drought: (a) ΔC at the pixel level; (b) total ΔC in each biome, where the asterisk indicates significant (*p* < 0.05). 1 Tg = 1000 Gg.

### 3.5 Long-term change in fire emissions

Global fire emissions increased in 1960–1994 while decreasing in 1995–2021. The Reanalysis of the Tropospheric chemical composition (RETRO) (Schultz et al., 2008) inventory dataset (1960–2000, obtained with the WebPlotDigitizer: https://automeris.io/WebPlotDigitizer/) was used to represent fire emissions in the 1960s–1990s because it was produced
with reliable inventory data and satellite fire detection. As shown in Fig. 8a, the RETRO data matched the proposed FiTCH data well. To detect breakpoints, annual fire emissions from the RETRO (1960–2000) and the FiTCH (2001–2021) were connected, producing long-term emissions from 1960 to 2021 (a synthesized time series). Using the R package '*segmented*' (Muggeo, 2008), 1994 was identified as the break year in the synthesized time series. Before 1994, fire emissions increased significantly at a rate of 27.09 Tg C yr$^{-1}$ (Fig. 8a). After 1994, the emissions decreased significantly by −25.90 Tg C per
275    year. Additionally, the mean annual global fire emissions after 1994 (2089.80 Tg C) were larger than before 1994, 2010.15 Tg C, indicating higher emissions in recent decades. In Fig. 8b, global annual fire emissions from the synthesized time series (RETRO + FiTCH) were positively correlated with global land temperatures before 1994, Spearman *r* = 0.42 and *p* = 0.013, as high temperatures contributed to more fire emissions. However, after 1994, the correlation became negative, Spearman *r* = −0.52 and *p* = 0.0053, indicating the decoupling of fire emissions and temperatures. In Fig. S3, the correlation between
temperatures and fire emissions in the RETRO was still positive and significant. However, the correlation in the FiTCH was not significant, which again indicated that fire emissions and temperatures were decoupled in the past two decades.

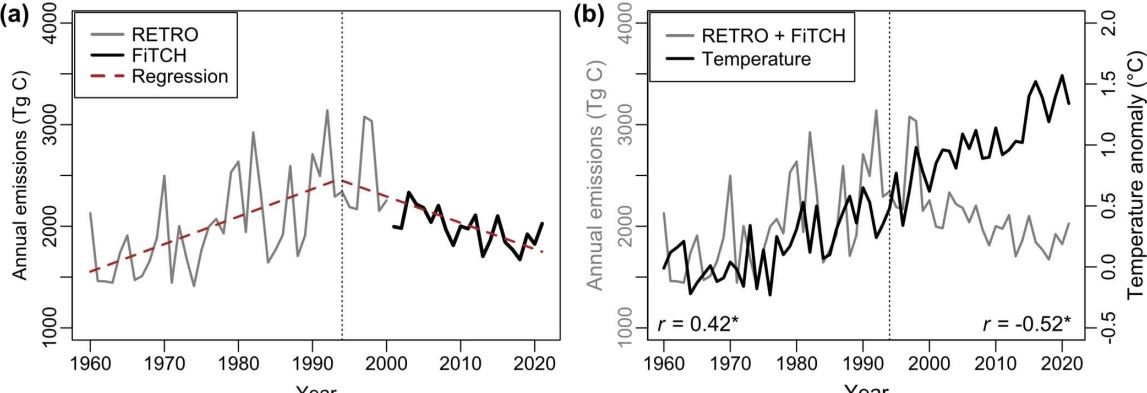

**Figure 8**. Synthesis of global fire emissions in 1960–2021: (a) time series of annual fire emissions, where the brown dashed lines are regression lines produced with the '*segmented*' package. The vertical black dotted line indicates the break year 1994; (b) long-term fire emissions (gray line) and global land temperature anomalies (black line).

## 4. Discussion

### 4.1 Uncertainties in fire emission products

Satellite derived fire carbon emissions were often used as references for global fire emission monitoring. However, there were large divergences among the current fire emission products. The means and the trends of annual emissions from the satellite-based fire emission products changed from 1978.47 to 3808.48 Tg C and −42.97 to −6.09 Tg C yr$^{-1}$ (Fig. 3), respectively. Additionally, the mean standard deviations (sd) of errors ranged from 4.99 to 0.78 Gg C (Fig. 1). Overall, the FiTCH, the GFAS, the GFED, and the QFED data presented low uncertainties (mean sd < 3.7 Gg C) and comparable fire emissions (~2000 Tg C per year). However, other products such as the FINN, the FEER, and Xu et al. (2021) data had high uncertainties (mean sd > 4.8 Gg C) and large fire emissions (2471.37 to 3808.47 Tg C per year).

These discrepancies were mainly due to the differences in models and data sources. The FINN data had the highest uncertainties among the fire emissions products, especially in Tropical and subtropical forests (TROPICS). This might be due to the fact that the FINN algorithm (version 2.5) estimated the burned area in dense forests (forest cover > 50%) with extended polygons (Wiedinmyer et al., 2023), which could make the estimated burned area larger than the satellite detected active fire regions. Thus, the burned area might be overestimated in tropical forests. Moreover, the fuel loads were designated based on land cover, which ignored the spatial variation of biomass in the same land cover. Tropical forests usually have high forest cover and the available fuel loads vary depending on species, soil, and water status. The errors in the burned area and fuel loads might result in high uncertainties in fire emissions. The FEER data also exhibited large

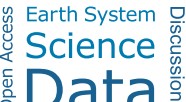

uncertainties in fire emissions. The emission coefficient ($C_e$, kg MJ$^{-1}$) in the FEER data was generated at a spatial resolution of 1° based on aerosol optical thickness (AOT), wind vectors, smoke aerosol mass extinction efficiency, and elevation. However, the smoke aerosol mass extinction efficiency was fixed for the whole globe (Ichoku and Ellison, 2014), which ignored its variation over land cover and climate regions. The coarse spatial resolution of $C_e$ also introduced errors when producing high spatial resolution data at 0.1°. Additionally, the potential of erroneously including surrounding aerosols from background environment to the focal fire plumes might cause the overestimation of $C_e$. The FEER data often produces

higher emissions than the GFAS and the GFED data (Ichoku and Ellison, 2014).

Xu et al. (2021) data presented the highest uncertainties in Boreal forests (Boreal) and Grasslands, savannas, and shrublands (GRASS) than other products (Fig. 2), which indicated the limitations of spaceborne LiDAR data in measuring burned biomass in both boreal forests and tree-sparse ecosystems. It is difficult to account for the consumed belowground biomass

using spaceborne LiDAR data, however, belowground biomass is often burned in boreal forest fires. Fig. S1 also revealed that there was underestimation of fire emissions in boreal regions when using the data from Xu et al. (2021). In tree-sparse ecosystems such as grasslands and savannas, the changes in canopy height and biomass due to fires were much smaller than those in forests, which caused challenges to accurately measure the consumed biomass using spaceborne LiDAR spare footprints. Other products such as the GFED, the GFAS, the QFED, and the proposed FiTCH also had some uncertainties,

however, their uncertainties were relatively small. The uncertainties due to models and data sources caused large variabilities among the satellite-based fire emissions. However, we still found that most products showed significantly decreased trends in fire emissions in the past two decades, which was consistent with the decline in the global burned area (Andela et al., 2017).

**4.2 Drought effects**

Drought is often related to fire because dry conditions dehydrate plants, reduce fuel moisture content, and make them burn easily. In this study, we derived the difference between fire emissions in drought years and non-drought years and found that the effects of drought on fire emissions had large heterogeneity. In forests such as Boreal forests, Temperate forests, and Tropical and subtropical forests (TROPICS), drought could lead to significant increases in fire emissions, where the increase in Boreal forests reached 242.09 Tg C yr$^{-1}$. In non-forest regions like Grasslands, savannas, and shrublands (GRASS),

drought caused negative effects on fire emissions, where the emissions decreased significantly. These differences corroborated that the effects of drought on fire emissions were multi-fold. For grass and shrub, their growth was constrained in drought years due to lack of water, and the biomass accumulation, i.e., fuel load, was smaller than that in normal years. In this scenario, the fire consumed biomass could be reduced, resulting in decreases in fire emissions. However, for forests, there was already a tremendous amount of biomass stored in stems, boles, and woody debris (van der Werf et al., 2017). Dry

conditions reduced fuel moisture and made trees more susceptible to fire, leading to increases in fire emissions in drought years. For boreal forests, the situation might be even worse as there was a large amount of belowground biomass which



could be burned during fire events. With the warming climate, hot and dry conditions would facilitate the consumption of belowground biomass in boreal regions.

### 4.3 Long-term fire emissions

The trends of long-term fire emissions varied with time. In 1960–2021, global annual fire emissions increased before 1994 but then decreased significantly. This result was consistent with the result from van Marle et al. (2017), where global fire emissions in 1750–2015 increased at the beginning, peaked in the 1990s, and then decreased gradually. The global burned area was also found to have decreased since 1998 (Andela et al., 2017). The global annual fire emissions were positively correlated with temperatures before 1994, while the correlation became negative after 1994 (Fig. 8b), which indicated the

decoupling of fire emissions and temperatures in the past two decades. The decreases in the burned area and fire emissions were mainly due to human activities such as agriculture expansion in Africa and deforestation reduction in tropical regions (Andela et al., 2017; Curtis et al., 2018), which were supported by the decreased fire emissions in Grasslands, savannas, and shrublands (GRASS) and Tropical and subtropical forests (TROPICS) biomes (Fig. 6).

Reducing deforestation in tropical forests is beneficial to forest carbon conservation and climate change mitigation. For environmental sustainability, it is urgent to decrease the forest deforestation rate globally. Unfortunately, deforestation in tropical forests has become a public concern again in the past several years, e.g., in 2019 and 2021. The increase in agricultural lands and land fragmentation in Africa also helped to constrain wildfires and decrease fire emissions since more lands were fragmented, regulated, and managed. Better land management and forest conservation strategies can diminish fire

risks and promote carbon sequestration. However, rising temperatures, VPD, and drought still cause substantial challenges to fire management. As is known, boreal forests and tundra store a large amount of belowground carbon, which is sensitive to temperatures and fire. Global warming causes more fire risks and carbon release in boreal forests (Phillips et al., 2022; Scholten et al., 2021), and increasing drought makes forests more susceptible to fire disturbances. More attention should be paid to forest fire management and fire control in the warming climate, especially in the northern high latitudes.

**5. Data availability**

The proposed FiTCH data are freely available from: https://doi.org/10.6084/m9.figshare.22647382.v1 (Liu and Yang, 2023).

The six satellite-based fire emission products were all from online resources. The GFED (van der Werf et al., 2017) data were from: https://www.globalfiredata.org/data.html; the GFAS (Kaiser et al., 2012) fire emissions were from :

https://ads.atmosphere.copernicus.eu/cdsapp#!/dataset/cams-global-fire-emissions-gfas?tab=overview;                    the            FINN (Wiedinmyer et al., 2023) fire emission data were from : https://rda.ucar.edu/datasets/ds312.9/index.html#!description; the FEER (Ichoku and Ellison, 2014) data were from: https://feer.gsfc.nasa.gov/data/emissions/; the QFED (Koster et al., 2015)



data were from: http://ftp.as.harvard.edu/gcgrid/data/ExtData/HEMCO/QFED/v2018-07/; fire emissions from Xu et al. (2021) were from: https://zenodo.org/record/4161694#.Y84tdnbMJm8.

## 6. Conclusions


This study analyzed the state-of-the-art fire emission products and developed a new global fire emission dataset for 2001–2021 using the TCH method, FiTCH. The uncertainties of different products were derived and illuminated. The proposed FiTCH dataset had the lowest uncertainties compared with the existing fire emission products, offering accurate global fire emission monitoring. Among biomes, Tropical and subtropical forests and Tundra had the highest uncertainties compared

with other biomes such as Temperate forests and Mediterranean forests. Additionally, biomes like Boreal forests revealed significantly increased fire emissions in the past two decades, which caused threats to ecosystem stability in northern high latitudes and climate change mitigation. Climate extremes like drought contributed to the increase of fire emissions in forests, making forests more vulnerable to the warming climate. Our results emphasized the importance of forest conservation in a changing world. Human activities, such as agriculture expansion, land fragmentation, deforestation

reduction, and fire suppression, have contributed to decreasing fire emissions in the past two decades. However, in recent years, e.g., in 2019 and 2021, deforestation has become a large concern again. Moreover, rising temperatures, increasing vapor pressure deficit, and growing drought risks cause great challenges to forest fire management and carbon mitigation. More attention should be paid to forest fire control and forest conservation in future warming climate for the prosperity of Earth's ecosystems.

## Author contribution


M.L. and L.Y. conceptualized the methodology and the framework. M.L. and L.Y. curated and processed the existing fire emission data. M.L. produced the new fire emission dataset, FiTCH. M.L. wrote the initial draft of the manuscript. L.Y. revised and improved the manuscript.

## Competing interests

The authors declare that they have no conflict of interest.

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
