# Peer review of "A global fire emission dataset using the three-corner hat method (FiTCH)"

_Earth System Science Data, 2023_

## Author Comment (AC1)

Referee 1:

This study compared six widely-used fire emission products and merged them using the three-cornered hat method (TCH). A new global fire emission dataset, FiTCH, was developed to quantify the fire emissions between 2001 and 2021. Fire emissions in different regions and biomes were derived and analyzed. The impact of drought on fire emissions was also evaluated. This study is timely and valuable as global climate change is expected to cause more frequent extreme events, such as extreme droughts and fires. Figuring out the uncertainties of the existing fire emission products and producing accurate fire emission data are important for global climate change analysis. Overall, this manuscript is solid and well-structured. However, the descriptions of some important points are inadequate. I suggest an accept after addressing the concerned and comments below.

Thanks for your valuable suggestions. We have improved our manuscript accordingly. Please refer to the point-to-point responses below.

Comments:

1. In Section 2.3, the impact of drought on fire emissions was quantified, which helped to illustrate the influence of drought on fire. Can the authors also analyze the effects of different drought severity? For example, -3 < PDSI < -2, -4< PDSI < -3, and PDSI < -4 usually indicate moderate drought, severe drought, and extreme drought, respectively. The fire emissions might change under different drought severity. Maybe add this extra analysis to the supplementary material.

We added a supplementary Figure S3 (shown below) to illustrate the effects of drought severity on fire emissions. Higher drought severity (e.g. extreme drought) did contribute to larger fire emissions in Boreal forests, TROPICS, Temperate forests, and Tundra. For GRASS, extreme drought caused larger decreases in fire emissions.

Please also refer to line 286 in the revised manuscript.

[Figure]

**Figure S3**. The impacts of drought severity (moderate, severe, and extreme drought) on fire emissions, where the asterisk indicates significant ($p < 0.05$). For extreme drought, the difference between average fire emissions ($\Delta C$) in extreme drought (PDSI < -4) years and non-drought years was calculated following the procedure in Eq. (8). The $\Delta C$ values for severe (-4< PDSI < -3) and moderate (-3 < PDSI < -2) droughts were also calculated using Eq. (8). Total $\Delta C$ was the sum of $\Delta C$ from the available 0.1° pixels. Some pixels might not have extreme drought years, and they were not considered when analyzing extreme drought. 1 Tg = 1000 Gg.

2. In Section 3.5, the correlations between fire emissions and temperature were described. However, only the RETRO data and the proposed FiTCH data were used. Can the authors also analyze the relationships between the six fire emission products and the temperature? For example, use a scatterplot for each product like Figure S3, with the x axis and the y axis for temperature and fire emissions, respectively. This analysis may also go to the supplement. Otherwise, Figure 8 will be too large.

We added the correlations between the six fire emission products and the temperature to supplementary Figure S4, which was shown below. The results were comparable to those shown in Figure 8, where fire emissions were decoupled from temperature (negatively correlated or not correlated) in the past two decades.

Please also refer to line 306 in the revised manuscript.

[Figure]

**Figure S4**. The correlation between temperature anomalies and annual fire emissions using the (a) RETRO (1960–2000), (b) FiTCH (2001–2021), (c) FINN (2002–2021), (d) FEER (2003–2021), (e) Xu et al. (2001–2019), (f) GFED (2001–2021), (g) QFED (2001–2021), and (h) GFAS (2003–2021). The blue lines are the regression lines (solid lines and dashed lines indicate p < 0.05 and p < 0.1, respectively).

3. The manuscript used the term "the three-corner hat", however, it is more common to use "the three-cornered hat". Maybe revise the term to make it consistent with the current literature.

We revised the term to "the three-cornered hat" throughout the manuscript. Please refer to line 1, line 10, line 72, and line 140 in the revised manuscript.

---

## Author Comment (AC2)

Referee 2:

Liu & Lang have assessed six different biomass burning emissions datasets using the three-cornered hat method. They constructed a new dataset and assessed relations between fires and climatological parameters. The paper reads well but I have fundamental doubts about the fidelity of the approach.

Thanks for your comments. We have improved our manuscript based on your comments. The major revisions include: 1) improved the descriptions of Section "2.1.1 Satellite-based fire emission products" to clarify the differences among the six fire emission products; 2) added a new Section "4.4. Caveats" to clarify the limitations of assuming the six fire emission products were independent; 3) added BB4CMIP data to the supplementary materials (Fig. S5).

Additionally, can you provide the "true values" frequently mentioned in your comments? The detailed data sources including downloading links, citations, and user manuals are necessary. It is amazing that you have true global fire emission data for the past two decades. We can re-train the TCH and re-evaluate the uncertainties of the six existing fire products if you provide the true global 0.1° fire emissions in 2001–2021.

Please refer to the point-to-point responses below.

As mentioned by the authors, the TCH method was developed to evaluate the uncertainties of different products when the true values were unknown. Each dataset is treated the same and considered independent. In reality that is not the case. GFAS is derived from GFED3 (which is rather similar as GFED4), and QFED builds on GFAS. Those datasets find their origin in the MODIS burned area data, which is also used by Xu et al. (2021). That means that four datasets share the same origin and FiTCH resembles those and yields low uncertainties to these four and relatively high to FEER and FINN which provide substantially higher emissions.

The four fire emission products: GFED4s, GFAS, QFED, and Xu et al. (2021) did not all rely on the MODIS burned area. We improved the descriptions of Section "2.1.1 Satellite-based fire emission products" to clarify the differences among the six fire emission products. Additionally, we added a new Section "4.4 Caveats" in the manuscript to clarify the limitations of assuming the six fire emission products were independent. The revisions were also shown below:

**2.1.1 Satellite-based fire emission products**

Six state-of-the-art satellite-based fire emission products were employed in this study. The GFAS data (version 1.2) (Kaiser et al., 2012) were based on the fire radiative power (FRP; Watt: W), which represented the released energy from biomass burning, from the MODIS active fire products rather than the MODIS burned area data. The FRP was integrated over time to produce the fire radiative energy (FRE; megajoule: MJ). Biome specific conversion factors (biomass consumed per megajoule; kg MJ$^{-1}$), which were derived from ground-based experiments and calibrated with the dry matter combustion rate in GFED (version 3; GFED3), were multiplied by the FRE to estimate the burned biomass and further derive the corresponding fire emissions. The 0.1° GFAS emission data for 2003–2021 were downloaded and analyzed in this study.

The latest version of the GFED data (version 4s, also called GFED4.1s) (van der Werf et al., 2017) were used in this study. The GFED4.1s used MODIS burned area and included burned area from small fires (smaller than a 500 m pixel) based on the statistical relationship between thermal anomalies, surface reflectance, and the burned area. Therefore, the burned area from the GFED4.1s was much higher than the MODIS burned area. The CASA model derived fuel loads and the corrected burned area were multiplied to estimate the burned biomass. The burned area and fire emissions from GFED4.1s were 37% and 11% higher than those from the GFED3 due to the inclusion of small fires. GFED4.1s data have been widely used by modeling communities, e.g. the Coupled Model Intercomparison Project Phase 6 (CMIP6; Van Marle et al., 2017). The GFED4.1s data in 2001–2021, whose original spatial resolution was 0.25°, were resized to 0.1° and employed in this study.

The latest QFED (version 2.5) (Koster et al., 2015) was based on the FRP and drew on the cloud correction method in the GFAS to produce continuous FRP observations over time. The fire emissions were estimated by multiplying the FRP and emission coefficients, which were the product of a constant conversion factor (1.37 kg $MJ^{-1}$), satellite factors, and biome specific scaling factors. The GFED global 0.1° fire emissions in 2001–2021 were employed in this study.

The spaceborne LiDAR data derived fire emissions in Xu et al. (2021) were produced with LiDAR based plant biomass and MODIS burned area. The carbon stocks, the MODIS burned area, and the emission factors were multiplied to produce global fire emissions, providing 0.1° annual fire emissions in 2001–2019.

The latest FINN fire emissions (version 2.5) (Wiedinmyer et al., 2023) used the bottom-up approach, in which fire emissions were derived from the product of the burned area and fuel loads. The burned area came from both MODIS and the Visible Infrared Imaging Radiometer Suite (VIIRS) active fire products, which were used to establish some extended or conservative polygons (i.e. burned area). The fuel loads were assigned based on the MODIS VCF product and MODIS land cover maps (MCD12Q1). The global 0.1° FINN data in 2002–2021 were used in this study. Additionally, to maintain the consistency of the fire emissions for the past two decades, MODIS based FINN data (FINN2.5 mod) were used here because VIIRS data were only available from 2012.

The FEER (version 1.0 G1.2) (Ichoku and Ellison, 2014) data were based on the total particulate matter (TPM), which was a product of the FRP and the estimated emission coefficients, which were based on wind vectors, smoke aerosol extinction efficiency, and aerosol optical depth. The TPM was converted to different components using emission factors from Andreae and Merlet (2001), producing a global 0.1° fire emission product since 2003. The FEER data in 2003–2021 were used in this study.

**4.4 Caveats**

When using the TCH method, the different data sources are usually assumed to be independent. However, this assumption might be too simple. In this study, the six fire emission products may have some kind of correlation due to the fact that they used some common data like the MODIS FRP and MODIS burned area. For example, GFAS, QFED, and FEER all employed the MODIS FRP to estimate fire emissions, though they had different strategies to adjust the FRP and emission coefficients. Both GFED4.1s and Xu et al. (2021) used MODIS burned area data, though GFED4.1s tried to correct the burned area by adding small fires.

We agree that the assumption of independent data sources is a limitation of this study. However, based on the BB4CMIP data, global fire emissions (10-year averages) varied between 1.8 and 2.3 Pg C yr$^{-1}$ (1 Pg = 1000 Tg), which were close to the proposed FiTCH dataset (1978.47 Tg C yr$^{-1}$). Additionally, the BB4CMIP showed that carbon emissions increased slightly and peaked during the 1990s, and after that they decreased gradually, which matched our results perfectly. We believe that our results are still reliable.

Now, in reality we do know to some degree what the true values are for biomass burning emissions. FEER uses top-down constraints and some recent work using Sentinel-2 burned area data indicates that MODIS severely underestimates burned area and the derived emission products thus severely underestimate emissions. See for example Ramo et al. (2021, PNAS, https://doi.org/10.1073/pnas.2011160118). Top-down studies also point towards higher emissions than calculated in the MODIS-based products, see for example Van der Velde et al. (2021) https://doi.org/10.1038/s41586-021-03712-y. Clearly these are regional studies but a wider look at the literature shows the same pattern; MODIS misses burned area.

The three-cornered hat method ignored this knowledge and considers the MODIS-based estimates as the best ones (I assume because they resemble each other) while in reality it is much more likely that FEER and FINN are closer to the truth.

For now, we do not have true fire emissions for the whole globe in the past two decades. Can you provide the true global 0.1° fire emissions from 2001 to 2021? Our study is trying to figure out the uncertainties of the existing products and produce a new product with low uncertainty for the past two decades. We can re-train the TCH and re-evaluate the uncertainties of the six fire emission products based on the "true values" you provide. Please provide the data sources you are referring to with detailed downloading links, citations, and user manuals.

We agree that the MODIS burned area data tend to underestimate burned area due to the coarse spatial resolution. However, GFED4.1s data corrected the MODIS burned area by including small fires, and the burned area increased by 37% at the global scale after adding small fires. Furthermore, GFAS, QFED, and FINN did not use the MODIS burned area data. GFAS and QFED used FRP, and FINN used active fire detection. Additionally, both Ramo et al. (2021) and Van der Velde et al. (2021) are just regional studies. We still do not know how much burned area was missing at the global scale when using the MODIS burned area data.

This research is trying to evaluate the uncertainties of the existing fire emission products when the true values are unknown rather than disrespecting or endorsing any fire products. We are not saying which product is the best or the worst because all the data have their own uncertainties and limitations. As we know, science is open-minded and welcomes diversity. All our efforts (the six existing products and the FITCH dataset) are to estimate and approach the true values. Please provide the "true values" if you have them. We can re-train the TCH and re-evaluate the uncertainties of the six fire emission products based on your "true values".

As a minor note, I would also be careful with using RETRO to go back in time. The uncertainties in the modelling and AVHRR approaches are very large and there has been more work done since that publication (BB4CMIP for example)

We added the BB4CMIP data to the supplementary Figure S5, which was also shown below. 1998 was identified as the break year, after which fire emissions decreased significantly. Before 1998, fire emissions from the BB4CMIP were positively correlated with temperature, Spearman $r = 0.34$ and $p = 0.031$, and the correlation decoupled after the break year. This result was comparable to Figure 8 in the manuscript.

Please also refer to line 309 in the revised manuscript.

[Figure]

**Figure S5**. Synthesis of global fire emissions in 1960–2021: (a) time series of annual fire emissions, where the brown dashed lines are regression lines produced with the 'segmented' package. The vertical black dotted line indicates the break year 1998; (b) long-term fire emissions (gray line) and global land temperature anomalies (black line). The BB4CMIP data in 1960–2000 were obtained with the WebPlotDigitizer (https://automeris.io/WebPlotDigitizer/).

In summary, I am sorry to say that I feel the hard work of the authors does not help the field forward. If 'true values' indeed would be unknown I would see the benefit of this work although there is still the issue that the products are not independent, but in fact there are hints towards the right magnitude of fire emissions in the recent literature and they point to the opposite conclusion of this paper.

For the "true values", can you provide the true wall-to-wall maps of global 0.1° fire emissions for the past two decades? We can re-train the TCH and re-evaluate the uncertainties of the six fire emission products based on the "true values" you provide. Please provide the data sources you are referring to with detailed downloading links, citations, and user manuals.

We agree that the assumption of independent data sources might be too simple. We improved the descriptions of Section "2.1.1 Satellite-based fire emission products" to clarify the differences among the six fire emission products, and we added a new Section "4.4 Caveats" to clarify the limitations of assuming the six fire emission products were independent.

Based on the six existing fire emission products and the FITCH dataset, the annual global fire carbon emissions between 2001 and 2021 ranged from 1978.47 to 3808.47 Tg C. The trends of global fire emissions in 2001–2021 were all decreasing, though the trends of GFED 4.1s and FINN were not significant. According to the BB4CMIP data you mentioned, global fire emissions (10-year averages) varied between 1.8 and 2.3 Pg C yr$^{-1}$ (1 Pg = 1000 Tg), which were close to the proposed FiTCH dataset (1978.47 Tg C yr$^{-1}$).

Additionally, the BB4CMIP showed that carbon emissions increased slightly and peaked during the 1990s, and after that they decreased gradually, which matched our results perfectly. We believe that the proposed FiTCH dataset captures the right magnitude and conclusions on global fire emissions from 2001 to 2021. We can also calculate the true magnitude of fire emissions and compare the "true values" with our FiTCH dataset if you provide the true well-to-well maps of global fire emissions between 2001 and 2021.

---

## Author Comment (AC4)

Referee #2

Nobody knows the true values of biomass burning emissions. However, in my review I have referred to recent literature that points out that MODIS burned area misses a lot of burned area and thus has a strong bias. Possibly a factor two, much more than the difference between GFED4s and GFED3. All derived emission products, also when based on FRP or active fires but tuned to datasets derived from MODIS burned area, have the same issue. This is becoming well established and accepted by the community and it means that global burned area and thus emissions are substantially higher than at least four of the datasets indicate.

The authors introduce a new dataset that very likely contains the same bias as GFED, GFAS etc because it is based solely on a statistical technique ignoring the new insights I point towards in my review. From a methodological perspective the 3CH method might be interesting, but in my opinion it does not help the field forwards. In contrast, it presents a new dataset that suffers from the same issue as some of the older ones but claims its uncertainty is very low.

A final note, I disapprove of the comment "Additionally, can you provide the "true values" frequently mentioned in your comments? The detailed data sources including downloading links, citations, and user manuals are necessary. It is amazing that you have true global fire emission data for the past two decades. ". I fully understand the frustration of a negative review but it is up to the editor to decide whether it is a fair review or not. In my review I have pointed towards pieces of information that are as close to reliable estimates as we can come at this stage and I feel it is our duty to build on that.

Thanks for the comments. We improved the newly added Section "4.4 Caveats", which is shown below, to clarify the limitations of this study.

**4.4 Caveats**

Global fire carbon emission data are crucial to fire dynamics monitoring, carbon cycling, and climate change mitigation. However, the true global fire carbon emissions are still unknown. This study tries to evaluate the uncertainties of the existing fire emission products, including GFAS, GFED4.1s, QFED, FINN, FEER, and Xu et al. (2021) data, when the true values are unavailable. The six state-of-the-art satellite-based fire emission products are analyzed and merged using the TCH method, producing a new global fire emission dataset, FiTCH.

When using the TCH method, the different data sources are usually assumed to be independent. However, this assumption might be oversimplified. In this study, the six fire emission products may have some kind of correlation due to the fact that they used some common data such as the MODIS FRP and MODIS burned area. For example, GFAS, QFED, and FEER all employed the MODIS FRP to estimate fire emissions, though they had different strategies to optimize the FRP and emission coefficients. Both GFED4.1s and Xu

et al. (2021) used the MODIS burned area data, though GFED4.1s corrected the burned area by adding small fires. We agree that the assumption of independent data sources is a limitation. However, based on the BB4CMIP data, global fire emissions (10-year averages) varied between 1.8 and 2.3 Pg C yr$^{-1}$ (1 Pg = 1000 Tg), which were close to the proposed FiTCH dataset (1978.47 Tg C yr$^{-1}$). Additionally, the BB4CMIP indicated that carbon emissions increased slightly and peaked during the 1990s, and after that they decreased gradually, which matched our results.

Due to the coarse spatial resolution of the MODIS data, small fires tend to be missed, resulting in an underestimation of the burned area and fire emissions. For example, at the global scale, the GFED4.1s produced 37% and 11% more burned area and fire emissions than the GFED3 due to the inclusion of small fires. Some regional studies (Ramo et al., 2021; van der Velde et al., 2021) also showed that the existing fire products underestimated the burned area and fire emissions. The proposed FiTCH dataset might have the same problem of underestimation as it still relies on these MODIS data-based fire products. However, the dilemma is that we do not know the true long-term fire carbon emissions at the global scale. Otherwise, the true uncertainties of these products can be quantified. This study is the first attempt to utilize the TCH method to assess the uncertainties of the six widely used fire emission products when the true values are unavailable. We agree that our results might not be perfect; however, they match the well-established dataset like the BB4CMIP. As we know, science is open-minded and welcomes diversity. We welcome more constructive ideas and suggestions about solving this problem.

References

Ramo, R., Roteta, E., Bistinas, I., Wees, D. van, Bastarrika, A., Chuvieco, E., and Werf, G. R. van der: African burned area and fire carbon emissions are strongly impacted by small fires undetected by coarse resolution satellite data, PNAS, 118, https://doi.org/10.1073/pnas.2011160118, 2021.

van der Velde, I. R., van der Werf, G. R., Houweling, S., Maasakkers, J. D., Borsdorff, T., Landgraf, J., Tol, P., van Kempen, T. A., van Hees, R., Hoogeveen, R., Veefkind, J. P., and Aben, I.: Vast CO2 release from Australian fires in 2019–2020 constrained by satellite, Nature, 597, 366–369, https://doi.org/10.1038/s41586-021-03712-y, 2021.